# UV-C LED Irradiation Reduces *Salmonella* on Chicken and Food Contact Surfaces

**DOI:** 10.3390/foods10071459

**Published:** 2021-06-24

**Authors:** Alexandra Calle, Mariana Fernandez, Brayan Montoya, Marcelo Schmidt, Jonathan Thompson

**Affiliations:** 1School of Veterinary Medicine, Texas Tech University, 7671 Evans Dr, Amarillo, TX 79106, USA; Mariana.fernandez@ttu.edu (M.F.); Marcelo.schmidt@ttu.edu (M.S.); 2Escuela de Medicina Veterinaria, Universidad Nacional de Costa Rica, Lagunilla, Heredia 40101, Costa Rica; brayan.montoya.torres@est.una.ac.cr; 3Department of Chemistry, Texas Tech University, MS 1061, Lubbock, TX 79409, USA; jon.thompson@ttu.edu

**Keywords:** UV-C, *Salmonella*, chicken, microbial intervention, food-contact surfaces

## Abstract

Ultraviolet (UV-C) light-emitting diode (LED) light at a wavelength of 250–280 nm was used to disinfect skinless chicken breast (CB), stainless steel (SS) and high-density polyethylene (HD) inoculated with *Salmonella enterica*. Irradiances of 2 mW/cm^2^ (50%) or 4 mW/cm^2^ (100%) were used to treat samples at different exposure times. Chicken samples had the lowest *Salmonella* reduction with 1.02 and 1.78 Log CFU/cm^2^ (*p* ≤ 0.05) after 60 and 900 s, respectively at 50% irradiance. Higher reductions on CB were obtained with 100% illumination after 900 s (>3.0 Log CFU/cm^2^). *Salmonella* on SS was reduced by 1.97 and 3.48 Log CFU/cm^2^ after 60 s of treatment with 50% and 100% irradiance, respectively. HD showed a lower decrease of *Salmonella*, but still statistically significant (*p* ≤ 0.05), with 1.25 and 1.77 Log CFU/cm^2^ destruction for 50 and 100% irradiance after 60 s, respectively. Longer exposure times of HD to UV-C yielded up to 99.999% (5.0 Log CFU/cm^2^) reduction of *Salmonella* with both irradiance levels. While UV-C LED treatment was found effective to control *Salmonella* on chicken and food contact surfaces, we propose three mechanisms contributing to reduced efficacy of disinfection: bacterial aggregation, harboring in food and work surface pores and light absorption by fluids associated with CB.

## 1. Introduction

*Salmonella* sp. is a major public health concern and a common food safety hazard associated with poultry processing [1,2,3,4]. Foodborne illness caused by this microorganism is one of the most frequent diseases affecting millions of people worldwide every year. Outbreaks related to *Salmonella* in poultry are very frequent [1,2,5]. A recent report by the Centers for Disease Control and Prevention (CDC), between 2015 and 2017, stated that poultry was associated with 262 outbreaks, 4807 illnesses, 849 hospitalizations and 12 deaths in the United States [6]. *Salmonella* is usually carried by live animals in their gastrointestinal track and transferred to processing environments where end-products can become contaminated [7,8]. Consistently, the presence of *Salmonella* in poultry houses is also very common with up to 100% prevalence among surveyed operations [9]. Efforts to control this pathogen are constantly made by the industry and government [10]. The most typical interventions to reduce *Salmonella* in poultry products involve the application of chemical treatments at different steps of processing, which include the use of organic acids, inorganic compounds, chlorine-based treatments, phosphate-based products, among other chemical compounds [11]. Consumers of poultry seem to have adverse opinions about the use of such chemicals in food [12], creating a challenge for the food industry to control bacterial contaminants. Therefore, poultry facilities would benefit from having alternative technologies to chemical interventions for pathogen control in food and processing environment.

The use of ultraviolet light has been proven to be effective for microbial inactivation by damaging bacterial DNA [13,14,15]. Pathogens absorb the ultraviolet (UV) light and thymine dimers are formed, blocking transcription and replication, which ultimately lead to cell death [15,16]. There is a growing interest in the use of UV treatments for the inactivation of pathogens in food [17]. The use of UV light in the food industry gained interest after the approval by the Food & Drug Administration (FDA) in 1997 to use UV irradiation as an alternative for microbial control in meat products [18,19]. Today, applications of UV light are commonly used to control pathogens in water, for decontamination of food contact surfaces (bakeries, dairy, and meat plants), and for decontamination of food packaging materials (boxes, bottles, leads, food wrapping films, thermoformable plastics, cartons for liquid foods and others) [19]. Commercially available equipment can be found advertised to disinfect surfaces, but most of the applications pertain to treating drinking water or being used for washing food products.

Several studies have demonstrated the effectiveness of using this technology in a wide variety of food products, such as fresh berries, apple juice, milk, fresh fish, processed meats, and in water [16,20,21,22,23,24]. Similarly, several studies have investigated the use UV light produced by mercury lamps, demonstrating the effectiveness of this technology in a wide variety of food products. However, mercury lamps require high voltage power supplies for operation, and certain lamps produce deep UV radiation of λ < 240 nm that generates significant quantities of ozone, a very reactive oxidative gas harmful to human health and food quality. UV light-emitting diodes (LED) are increasingly being used as substitutes for mercury lamps for several reasons. UV LEDs are much smaller than mercury lamps and generate less heat. As a result, they may be placed close to food contact surfaces to achieve high irradiance, and presumably more effective inactivation of pathogens. In addition, the emission spectrum of UV LEDs can be tuned to emit UV light specifically of wavelengths between 250–280 nm, which are most effective at driving the photochemical reactions leading to formation of thymine dimers. Considering the need to control *Salmonella* in poultry operations, this research aimed to evaluate the effectiveness of UV-C LED light for the reduction of *Salmonella* sp. applied to the surface of chicken breasts (CB), stainless steel (SS), and high-density polyethylene (HD) using different times and irradiance intensities.

## 2. Materials and Methods

### 2.1. Bacterial Cultures

A five-strain *Salmonella* cocktail was prepared with *Salmonella* Thyphimurium ATCC BAA-712, *Salmonella* Newport ATCC 6962 (food poisoning fatality), *Salmonella* Enteritidis ATCC 31194, *Salmonella* Senftenberg ATCC 43845, and *Salmonella* Heidelberg ATCC 8326. Each strain was grown individually by transferring 10 μL from the stock culture into 9-mL of Tryptic Soy Broth (TSB) (EMD Millipore Chemicals; Darmstadt, Germany) and incubating for 24 h at 37 °C. Equal amounts (2 mL) from each grown *Salmonella* suspension were combined into a sterile test tube and homogenized. The bacterial cocktail was freshly prepared prior each repetition. *Salmonella* concentration in the cocktail was confirmed at each repetition of the experiment by conducting serial dilutions and plating onto Trypticase Soy Agar (TSA) (Becton, Dickinson and Company, Franklin Lakes, NJ, USA), followed by incubation for 24 h at 37 °C and subsequent enumeration.

### 2.2. UV-C LED Light and Surfaces Subjected to Irradiation

The ultraviolet type C (UV-C) light used as the irradiation source for this project was a Klaran class LED acquired from Crystal IS Inc. (Green Island, NY, USA). The UV-C LED had a wavelength range of 250–280 nm, 20 mW power and a viewing angle of 105 degrees. The lamp was operated under forward bias at a maximum 400 mA current, corresponding to 100% irradiance, which is the maximum current recommended by the manufacturer. The average irradiance used in this study was either 2 mW/cm^2^ (referred in this experiment as 50% or half irradiance) or 4 mW/cm^2^ (referred in this experiment as 100% or full irradiance). Three different surfaces were treated with UV-C LED irradiation: (1) boneless skinless chicken breast (CB), (2) stainless steel (SS) and (3) high density polyethylene (HD). To treat each surface, experiments were carried out on 2 × 2 cm coupons used as the experimental units. SS and HD were selected to be treated with the UV-C light since they are commonly used as food-contact surfaces in the poultry processing industry. 

### 2.3. Chicken Inoculation and Treatment

Chicken breast was obtained boneless and skinless from a local supermarket. Portions of 2 × 2 cm and approximately 4 mm thick were aseptically cut. The upper surface was inoculated with the five-strain *Salmonella* cocktail at a target concentration of ca. 6.0 Log CFU/cm^2^. The inoculated CB squares were placed on a tray and set under refrigeration for 30 min to allow for bacterial attachment. Two irradiance conditions, 50 and 100%, were explored. In all cases, the CB squares were irradiated individually under the UV-C LED source. In the first case (50% irradiance), the CB squares were treated for varying times with integrated doses of UV-C radiation corresponding to 0–1.8 J/cm^2^. For the second treatment (100% irradiance), the UV-C dose ranged from 0–3.6 J/cm^2^. As light intensity scales linearly with drive current, the UV-C irradiance was controlled by metering the drive current of the LED. The exposure times were: 60, 180, 300, 600 and 900 s. An additional control set of samples (inoculated, not irradiated) were considered. Control samples are referred as 0 s. 

### 2.4. Stainless Steel Inoculation and Treatments 

Stainless Steel 304 (SS, C 0.08% max., Mn 2.00% max., P 0.045% max., S 0.03% max., Si 0.75% max., Cr 18.00–20.00%, Ni 8.00–12.00%, N 0.10 max., Fe balance), 2 mm thickness was obtained from Agrosuper (Rancagua, Chile). Sterile SS squares were surface inoculated before each experiment. Squares were cleaned, degreased with acetone, flamed with 95% ethanol, stored in a glass container and autoclaved at 121 °C (15 lb/in^2^) for 15 min. Sterile SS squares were surface-inoculated by applying a 20 µL aliquot of the five-strain *Salmonella* cocktail on one side of each 4 cm^2^ square. A target surface inoculation of 6.5 Log_10_ CFU/cm^2^ was attempted. The inoculum was completely spread on the entire surface using a sterile 1-µL loop and then let sit for 30 min under refrigeration to dry and for bacterial attachment. Treatments were performed with both the low and high irradiance cases, applying a spatially averaged irradiance of approximately 2 mW/cm^2^ and 4 mW/cm^2^, respectively. Irradiation occurred for a period of 15, 30, 45 and 60 s, and additionally for a control set of samples (inoculated, not irradiated). Controls are referred to as 0 s.

### 2.5. High Density Polyethylene Inoculation and Treatments

Kitchen cutting boards (approx. 1 cm thick) were obtained from the microbiology research lab, which had been previously used to chop meat samples. The cutting boards were intentionally chosen as used to mimic scratched surfaces from processing facilities. Prior to the study, the HD board was cut into 2 × 2 cm squares (4 cm^2^). HD squares were treated and inoculated following the same procedures as with SS. Both the full irradiance (100%) and half irradiance (50%) cases were considered. Irradiation times included trials for: 30, 60, 90, 120, 150, 180, 300, 600 and 900 s. Additionally, control samples (inoculated, not irradiated) were tested and referred as 0 s. 

### 2.6. Analysis of Chicken Rinse Fluid 

Fluids associated with CB were analyzed to evaluate whether they could offer a protective coat effect for bacteria by absorbing ultraviolet light. The extent of light absorption by CB juices was studied by ultraviolet-visible (UV-VIS) absorption spectroscopy. A portion of chicken breast was placed in a plastic bag and 10 mL of deionized water was added to wash the surface of the chicken. A 3 mL portion of the deionized water was collected and placed into a 1 cm path length quartz cuvette, and the full UV—VIS absorbance spectrum was recorded against a deionized water blank on an Agilent photodiode array spectrometer with 1 nm spectral resolution.

### 2.7. Microbial Analysis 

CB portions were placed immediately after the treatment into 9-mL Buffered Peptone Water (BPW) (BD BBL™, Franklin Lakes, NJ, USA) tubes and thoroughly homogenized. Serial dilutions were conducted to facilitate enumeration followed by spread plating on Xylose-Lysine-Tergitol 4 (XLT4) (BD Difco™ Franklin Lakes, NJ, USA). Inoculated XLT4 plates were incubated for 24 h at 37 °C. SS and HD squares exposed to the LED UV-C treatment were transferred immediately after the exposure time to sterile conical tubes (50 mL capacity, Corning™ Falcon™) containing 10 mL of phosphate-buffered saline solution (PBS, Sigma-Aldrich^®^, Saint Louis, MO, USA), then mixed by vortex motion for 60 s to transfer the bacterial cells from the surface to the saline solution. The number of viable bacteria in the saline solution was determined by serially diluting with BPW, spread plating on XLT4 plates and incubating for 24 h at 37 °C. For each surface, colonies were enumerated upon incubation, and final counts were reported as CFU/cm^2^ considering the size of CB, SS, and HD coupons of 4 cm^2^. Control samples were also enumerated following the corresponding protocol. 

### 2.8. Electron Micrographs

Scanning electron microscope (SEM) images were taken by the Texas Tech College of Arts and Sciences Microscopy (CASM). Samples were provided to CASM frozen at −80 °C with the bacterial cells suspended in sterile water. CB, SS, and HD squares with bacterial cells were dried frozen and coated with Iridium (Ir). SEM imaging were obtained with an electron microscope Zeiss Crossbeam 540 FIB-SEM.

### 2.9. Statistical Analysis 

Each surface (CB, SS and HD) treated with the UV-C was subjected to two different treatment combinations that included irradiance and exposure time. Analyses of variance were used to test the effect of time periods (illumination time; measured in seconds) on *Salmonella* reduction (Log CFU/cm^2^) under two levels of irradiance exposure (irradiance), and on three specific surfaces conditions (i.e., chicken breast, stainless steel, and high-density polyethylene. Three experimental repetitions were conducted and a total of six separate ANOVAs were conducted. Each model revealed a significant (α = 0.05) UV-C illumination time effect on Log CFU/cm^2^. Multiple comparisons were calculated using Bonferronni correction to determine differences at each level of the illumination time variable. All statistical analyses were conducted with STATA (StataCorp. 2019. Stata Statistical Software: Release 16. College Station, TX, USA: StataCorp LLC.).

## 3. Results and Discussion

LED UV-C treatment was applied to inactivate *Salmonella* sp. deposited on three different surfaces: chicken breast (CB), type 304 stainless steel (SS) and high-density polyethylene (HD). For all samples tested, two irradiance intensities were tested, 2 mW/cm^2^ (50%) and 4 mW/cm^2^ (100%). Illumination times between 0 and 900 s (0 and 15 min) were explored. An overview of the findings per treatment is summarized in Table 1, Table 2 and Table 3 and discussed below. The UV-C wavelengths used during the experiments were in the range 250–280 nm, which are considered safe for food products according to the FDA permitted levels of 253.7 nm [25]; however, this regulation only refers to the use of mercury lamps and not LED lamps. 

### 3.1. Boneless Skinless Chicken Breast (CB)

Results for reduction of *Salmonella* on CB are reported in Table 1. For the CB treated with 50% irradiance, initial bacterial attachment was estimated to be 6.21 ± 0.16 Log CFU/cm^2^. Significant (*p* ≤ 0.05) reductions of *Salmonella* were obtained after each of the treatment times (60, 180, 300, 600 and 900 s) compared to the starting inoculation level. After 60 s of exposure, *Salmonella* decreased by 1.02 Log CFU/cm^2^, which was significant at *p* ≤ 0.05. Upon completion of a 900 s irradiance, a total *Salmonella* reduction of 1.78 Log CFU/cm^2^ (*p* ≤ 0.05) was achieved. 

On the other hand, when CB was treated with 100% irradiance the reduction of *Salmonella* was enhanced. Considering the initial attachment level of *Salmonella* observed in the samples (6.26 ± 0.11 Log CFU/cm^2^), significant (*p* ≤ 0.05) reductions were also obtained after each treatment time relative to the *Salmonella* level before treatments. Data show a total reduction of >3.0 Log CFU/cm^2^ during the total exposure time (900 s). Based on the data obtained at the different time points, the rate of reduction of *Salmonella* occurred most efficiently within the first 60 s of UV illumination. During this time, *Salmonella* was reduced by 2.05 Log CFU/cm^2^, which was significant at *p* ≤ 0.05. After that first minute of UV-C exposure, *Salmonella* was reduced only by an additional 0.96 Log CFU/cm^2^ total, which was still significant at *p* ≤ 0.05. Comparable results were found by McLeod et al. [26]. In their investigation using 254 nm wavelength, skinless chicken fillets were exposed for 5, 10, 30, 60 and 300 s. After the first 60 s of treatment, they were able to observe a *Salmonella* reduction of 1.5 Log CFU/cm^2^. However, when the exposure was 300 s, a 2.4 Log CFU/cm^2^ reduction was achieved. 

There are three hypothesis we propose for this outcome. First, the porosity of the chicken surface could play an important role in the reduced effect of the UV-C against *Salmonella*. An image of the boneless skinless chicken breast sample used during this research was obtained via electron microscopy (Figure 1). The chicken surface may visually appear smooth; however, cracks, crevices, and/or pores could protect bacterial cells from light exposure [26]. A single *Salmonella* cell is 2–5 µm long by 0.5–1.5 µm wide [27], thus the micro holes in the chicken breast, as observed in the electron micrograph (Figure 1A), could harbor bacterial cells. Some cells may become trapped or sequestered within the irregular and porous surface of the chicken, which may affect the effectiveness of the UV-C to evenly cover and reach the entire surface area of the sample as seen in Figure 1B,C. As described by Lagunas-Solar et al. [28], complex surface properties of foods bring a challenge; microorganisms located in pores and crevices of a food surface can be shaded from light, and thus remain unaffected. The use of UV light can be more effective to reduce microorganisms on foods with smooth surfaces such as fresh whole fruits, vegetables, hard cheese, and smooth-surface meat slices [29].

The second hypothesis for reduced efficacy of UV illumination after the initial minute is that the fluid on the surface associated with CB absorbs ultraviolet radiation, reducing the light intensity and thereby reducing the rate of bacterial deactivation. To support this premise, the UV-VIS absorption spectrum of the fluid associated with the chicken breast samples was obtained. It was found that the fluid strongly absorbs ultraviolet light below 300 nm, and this light absorption will lower the intensity of light interacting with *Salmonella*, offering a protective or shielding effect to prevent deactivation (see Section 3.5). The presence of fluids between bacterial cells and the light, most likely affects the efficacy of the treatment as the liquid may absorb the light [26,30]. 

The third hypothesis involves the tendency for cells to aggregate into clusters. When illuminated, cells near the surface of the cluster (nearest to LED) may absorb the UV radiation and be inactivated. However, cells located beneath the top layer may be shaded from full illumination and protected. The premise for this mechanism of bacterial cell protection during UV irradiation has been presented recently in the literature [31]. 

### 3.2. Stainless Steel (SS) 

The reduction of *Salmonella* was evaluated at intervals of 15 s during a period of 60 s, and the summary of the findings is presented in Table 2. For experiments with 50% irradiance, the initial attachment level of *Salmonella* on the SS squares was only 3.4 ± 0.61 Log CFU/cm^2^. Results indicate that a rapid reduction (1.3 Log CFU/cm^2^) of *Salmonella* occurred after the first 15 s of exposure to UV light, which was statistically different (*p* ≤ 0.05) from the starting level. When the total exposure time of 60 s was applied, a reduction of 1.97 Log CFU/cm^2^ (*p* ≤ 0.05) was observed.

In the case of experiments with 100% irradiance, SS was inoculated with an average load of 6.26 ± 0.49 Log CFU/cm^2^ of *Salmonella*. A loss of approx. 1.36 Log CFU/cm^2^
*Salmonella* occurred within the first 15 s of illumination, and nearly 2.49 Log CFU/cm^2^ was reduced after 30 s of exposure to the UV LED. These reductions were statistically significant at *p* ≤ 0.05. The reduction of *Salmonella* over time on SS did not plateau and level-off, but rather a decrease in numbers continued through the entire duration of the experiment. The highest reduction was observed after 60 s of UV-C exposure (3.48 Log CFU/cm^2^). Lim and Harrison [14] evaluated the effect of UV-C (254 nm) in reducing *Salmonella* contamination on 3 × 5 cm stainless steel coupons. They obtained reductions of 2.75 and 3.51 Log CFU/coupon of 15 cm^2^ after treatment times of 5 and 30 s, respectively. Bae and Lee [32] exposed stainless steel for longer periods and found reductions of 1.25 and 2.02 Log CFU/coupon of 5 × 2 cm after 30 min and 1 h, respectively. While it appears that their investigation suggests a low effectiveness of the UV treatment, it is important to mention that their group used a UV 253.7 nm wavelength with intensity of 0.236 ± 0.013 mW/cm^2^, which was much lower than the irradiance used in the current research (2 or 4 mW/cm^2^). Consistent reductions were also observed by Sommers et al. [30]. Their findings indicate a *Salmonella* reduction of 5 Log CFU/coupon on stainless steel when inoculated coupons were exposed to UV-C at a dose of 400 mJ/cm^2^. When inoculated coupons were treated with 50 mJ/cm^2^, the pathogens were reduced by only 1.86–3.05 Log CFU/coupon. Kim et al. [31] found that UV-C intensities of 250 or 500 μW/cm^2^ decreased three target microorganisms (*L. monocytogenes*, *S. typhimurium*, and *E. coli* O157:H7) on stainless steel surfaces. A UV-C dose of 90 mJ/cm^2^ reduced the three pathogens by >4 Log CFU/coupon; however, a dose of 15 mJ/cm^2^ decreased the pathogens by 2.43–4.38 Log CFU/sample. These doses and times were considerably higher (1, 2, and 3 min) compared to those use in the current research.

Based on the above cited investigations, it may be possible to increase the rate of *Salmonella* destruction by increasing exposure times. To investigate whether the porosity of the SS surface causes harboring of cells, electron micrographs of the SS coupons used during the present experiment were obtained (Figure 2). The images show minor surface imperfections. Although the depth can’t be determined, the apparent size of the crevice may not be large enough to harbor *Salmonella* cells (Figure 2A). An agglomeration of cells was also observed (Figure 2B), forming horizontal and vertical layers of cells (Figure 2C).

### 3.3. High Density Polyethylene (HD)

*Salmonella* reduction was observed when HD was treated with UV-C, as presented in Table 3. For the treatment of HD with 50% irradiance, the initial inoculation level was 6.58 ± 0.16 Log CFU/cm^2^. After 30 s of exposure, *Salmonella* was reduced by nearly 1 Log CFU/cm^2^ (*p* > 0.05); however, only after 150 s of irradiation was a significant (*p* ≤ 0.05) reduction in *Salmonella* obtained. Disinfection on the HD surface followed a different temporal pattern compared to CB, as statistically significant reduction of *Salmonella* continued to be achieved after even several minutes of illumination. This result suggests the *Salmonella* on HD surfaces may not experience the shielding effect proposed for the CB samples (*vide supra*). 

When HD squares were treated with 100% irradiance, *Salmonella* was also effectively reduced. The initial attachment level of the microorganism was 5.20 ± 0.15 Log CFU/cm^2^. Experimental data showed a statistically significant (*p* ≤ 0.05) reduction of 1.77 Log CFU/cm^2^ during the first 60 s of exposure, with approximately 1.23 Log CFU/cm^2^ reduction (*p* > 0.05) occurring within the initial 30 s. Lim and Harrison [14] obtained similar results when they exposed 35 cm high density polyethylene coupons inoculated with *Salmonella*. After 5 and 30 s of treatment with UV-C light (254 nm), the reduction of *Salmonella* was 2.93 and 4.32 Log CFU/coupon of 15 cm^2^, respectively. In 2011, Haughton et al. [33] treated nine different food contact surfaces (black & white polypropylene, polystyrene, aluminum, polyethylene-polypropylene blend, polyolefin, polyvinyl chloride, stainless steel, polyethylene) with UV-C does ranging from 0–192 mJ/cm^2^. The authors found that *C. jejuni*, *E. coli*, and *Salmonella* could be reduced by >2 Log CFU/cm^2^ on all surfaces during treatment. However, substantial differences in disinfection efficacy were noted for different materials. For the polyethylene cutting board tested, a UV-C dose of <20 mJ/cm^2^ was effective at inactivating the bacteria to levels below the limit of detection. Bae and Lee [32] obtained reductions of 1.62 and 1.18 Log CFU/coupon of 5 × 2 cm after 30 min and 1 h of UV treatment. Although the authors reported statistically significant reductions relative to the level of *Salmonella* before treatments, they were considerably lower than the reductions found in the present research. 

Longer exposure times yielded much higher inactivation levels of *Salmonella*. After 180 s of treatment, no *Salmonella* was recovered from the samples in any of the repetitions. To confirm the inactivation of *Salmonella*, samples exposed to the UV-C treatment during 180, 300, 600 and 900 s were enriched in 10-mL BPW, incubated overnight at 37 °C, and streaked onto XLT4. After 24 h of incubation at 37 °C no *Salmonella* colonies were recovered. Sommers, et al. [34] inoculated both stainless steel and HDPE surfaces with *F. tularensis* in food exudate prior to treating with UV-C. These authors found that exposure to 500 mJ/cm^2^ reduced the pathogen level by >4 Log CFU/coupon for both surfaces. However, their treatment was at a higher UV dose, and it is possible that *F. tularensis* is less sensitive to UV treatment compared to *Salmonella*. 

Electron micrograph of the HD coupons used during these experiments were obtained. As depicted in Figure 3, deep crevices were observed (Figure 3A,B), which could potentially hide bacterial cells. These crevices may be associated with use of the board for cutting. Clumping of cells was observed (Figure 3C). Thus, reductions obtained when the UV dose was >0.72 J/cm^2^, indicate that the quality of the surface may not have affected bacterial survival. 

Only in the case of HD did the UV dose seem to show consistent reductions regardless of what combinations of irradiance and time were used to achieve the given dose. For example, when the UV dose was 0.12 J/cm^2^, a reduction of 1.25 and 1.23 Log CFU/cm^2^ was observed at 50 and 100% illumination, respectively. Similar cases were observed with 0.24, 0.36, and 1.2 J/cm^2^ as observed in Table 3. This appears to follow the Bunsen–Roscoe reciprocity law, which suggests that the effectiveness of the irradiation is achieved regardless of what combination of time and irradiation rate is used to reach a certain UV dose exposure (short exposure with high irradiance or long exposure time with high irradiance) [30].

### 3.4. Cell Clumping and SEM Images 

SEM images obtained from the inoculated surfaces and chicken are presented in Figure 1, Figure 2 and Figure 3. On CB and HD, pores were large enough to shelter *Salmonella* cells. Electron micrographs of SS show clear scratches that are long but apparently not wide or deep enough to harbor *Salmonella*. As an important finding, SEM images with bacterial cells inoculated on the CB, SS and HD show vertical and horizontal accumulations of cells (clumping or aggregation). The conglomeration of cells may also cause shading and shielding effects, protecting those cells that are below the top layer, as previously mentioned [31]. It can be hypothesized that with a larger concentration of cells on the surfaces, UV-C light penetration could represent a challenge. This possibility should consider the fact that chicken and food contact surfaces could also carry other microorganisms that could potentially cause shielding. 

### 3.5. Absorption of UV Light by Chicken Rinse Fluid 

An extremely high absorption of light by the fluid present on the chicken at wavelengths lower than 300 nm was observed. The resulting absorbance spectrum is depicted in Figure 4. The data suggests that <0.01% of light below 290 nm was transmitted through the 1 cm path sample used. The fluid associated with the chicken breast absorbs UV light very strongly, and bacteria immersed within this fluid are likely protected or sheltered from photochemical damage caused by irradiation by the LED. This effect may cause the observed rapid initial reduction in bacterial load followed by leveling off between 4–5 Log CFU/cm^2^. Bacterial cells not well immersed within the fluid may experience full illumination from the LED and resultant deactivation, while other bacterial cells more immersed within the fluid/broth are sheltered by the fluid’s absorption of light and are protected. 

The effectiveness of UV-C light on disinfecting liquids is known to be dependent on the type of fluid [35]. A low transmittance of UV light is common in fluids other than water due to their tendency to scatter and/or absorb UV light [12]. When liquids have low transmissivity due to the presence of organic compounds, soluble solutes or particulate matter, UV-C disinfection can be challenging [36]. As a point of reference, the penetration depth of some fluid foods (the distance at which 90% of the light is absorbed) is 0.67, 0.25, 0.22, 0.10 and 0.01 mm for clear apple cider, apple cider, liquid sucrose, orange juice, and egg whites, respectively [12]. 

The commercial availability of deep UV-C LEDs has led to an emergence of potential applications in the food processing industry [37,38]. Due to their advantages, LED lamps are now being implemented in systems for water disinfection; however, other uses are currently rare. One exciting application is in the disinfection of food products and food contact surfaces while on the production line. LED devices could be more robust, durable, and portable compared to mercury lamps because there are no glass tubes that may break and contaminate workstations with mercury. 

## 4. Conclusions

This study demonstrated the effectiveness of UV-C LED at reducing *Salmonella* on chicken breast samples and common food contact surfaces such as stainless steel and high-density polyethylene. At a minimum, a 1 Log CFU/cm^2^ reduction for CB was noted in trials, with up to 3 Log CFU/cm^2^ being reached. Further reductions seemed to be limited by the remaining *Salmonella* in the sample being shaded from the UV-C light. This is believed to occur by *Salmonella* sheltering within pores on the CB surface or behind neighboring bacterial cells, absorption of UV light by fluid present on the CB, or both effects simultaneously. *Salmonella* was also reduced on both food contact surfaces, yielding reductions up to 3.5 and 5.2 Log CFU/cm^2^ on stainless steel and high-density polyethylene, respectively. An increase in irradiance yielded higher reductions of *Salmonella* on food and food contact surfaces with up to 99.999% in the case of HD.

A clumping cell factor, when large number of bacteria are accumulated on the surfaces, should be considered. Electron micrographs showed formation of layers of *Salmonella* that extended horizontally and accumulated vertically, which could protect cells beneath the top layer. 

UV-C LED illumination could be an effective means to deactivate *Salmonella*, especially for nonporous surfaces which are not UV light absorbing. 

By doubling the irradiance (mW/cm^2^) from 50 to 100%, the UV dose (J/cm^2^) deposited on each surface was also increased or duplicated. Larger UV doses were directly correlated with the *Salmonella* reduction (Log CFU/cm^2^) attained on each surface tested; however, such reduction did not necessarily double. In other words, *Salmonella* reductions were consistent with the intensity of exposure but not exactly proportional to the increase in the UV dose.

The majority of research studies investigating the effect of UV treatments to control bacterial pathogens from food or food contact surfaces focus on the use of conventional mercury UV lamps. Since the present investigation found the effectiveness of using UV-C LED light for food and environmental surface treatment, findings could be relevant particularly to the poultry industry. The advantages of UV-C LEDs over chemical treatments and conventional mercury UV should be highlighted when considering UV-C LEDs as an alternative for pathogen control. UV-C LEDs do not contain mercury, are environmentally friendly, robust, durable, energy efficient, and their full illumination power can be reached more rapidly, without time delay for warm-up [39]. 

## Figures and Tables

**Figure 1 foods-10-01459-f001:**
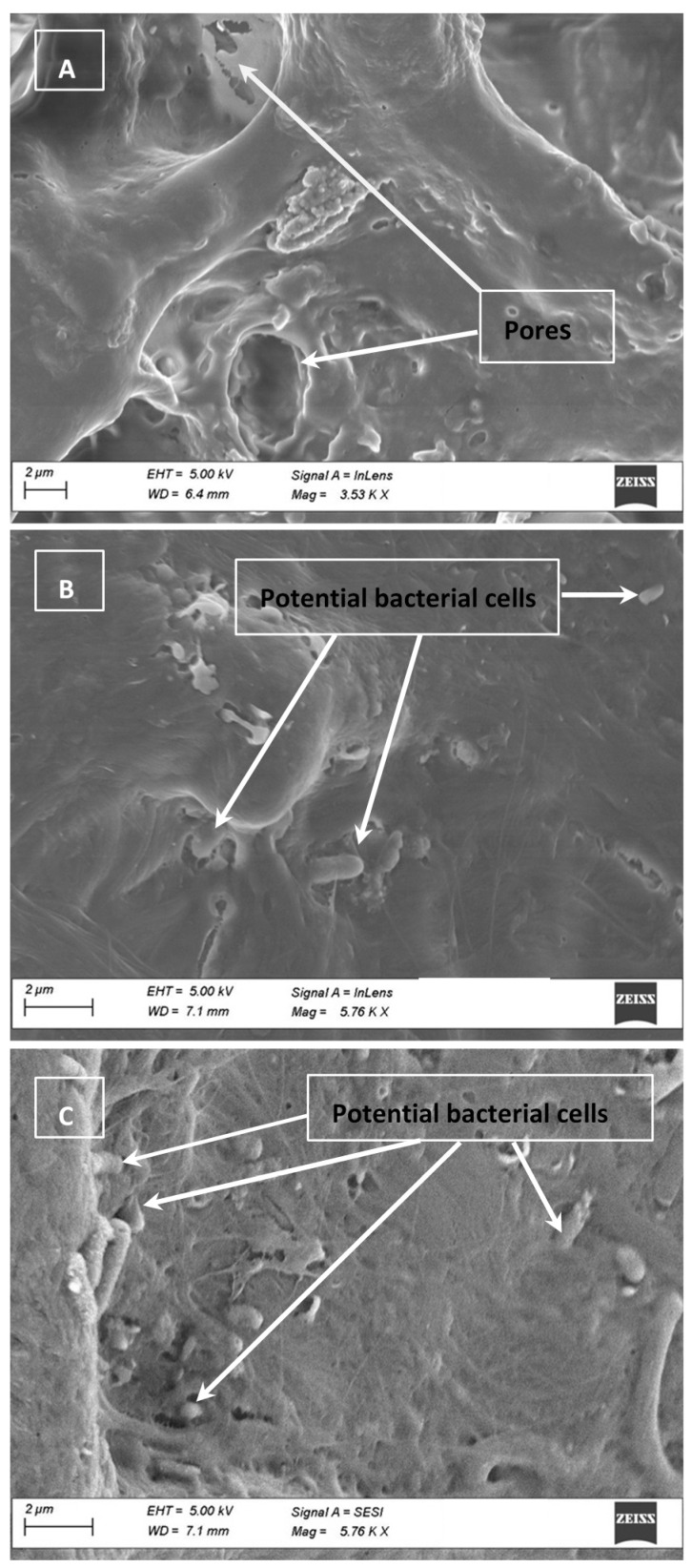
Electron micrograph of chicken breast samples illustrating the porous nature of the chicken (**A**). *Salmonella*, being only 2–5 µm long by 0.5–1.5 µm wide may enter pores on the surface of the chicken and be sheltered from full illumination (see (**B**,**C**)).

**Figure 2 foods-10-01459-f002:**
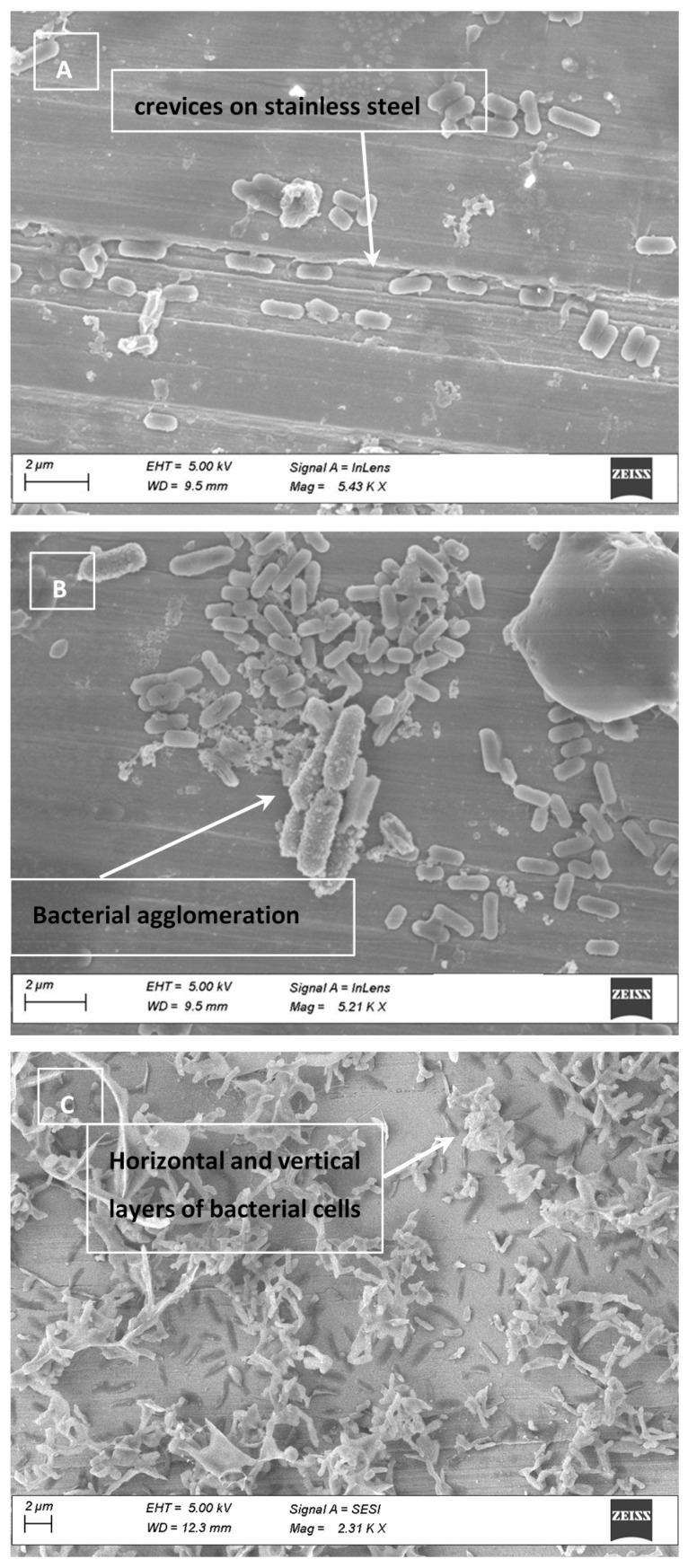
Electron micrograph of stainless steel (SS) inoculated with *Salmonella* showing imperfections on SS (**A**) and agglomeration of the cells on the surface (**B**). Vertical and horizontal agglomerations were observed when high volumes of cells are present (**C**).

**Figure 3 foods-10-01459-f003:**
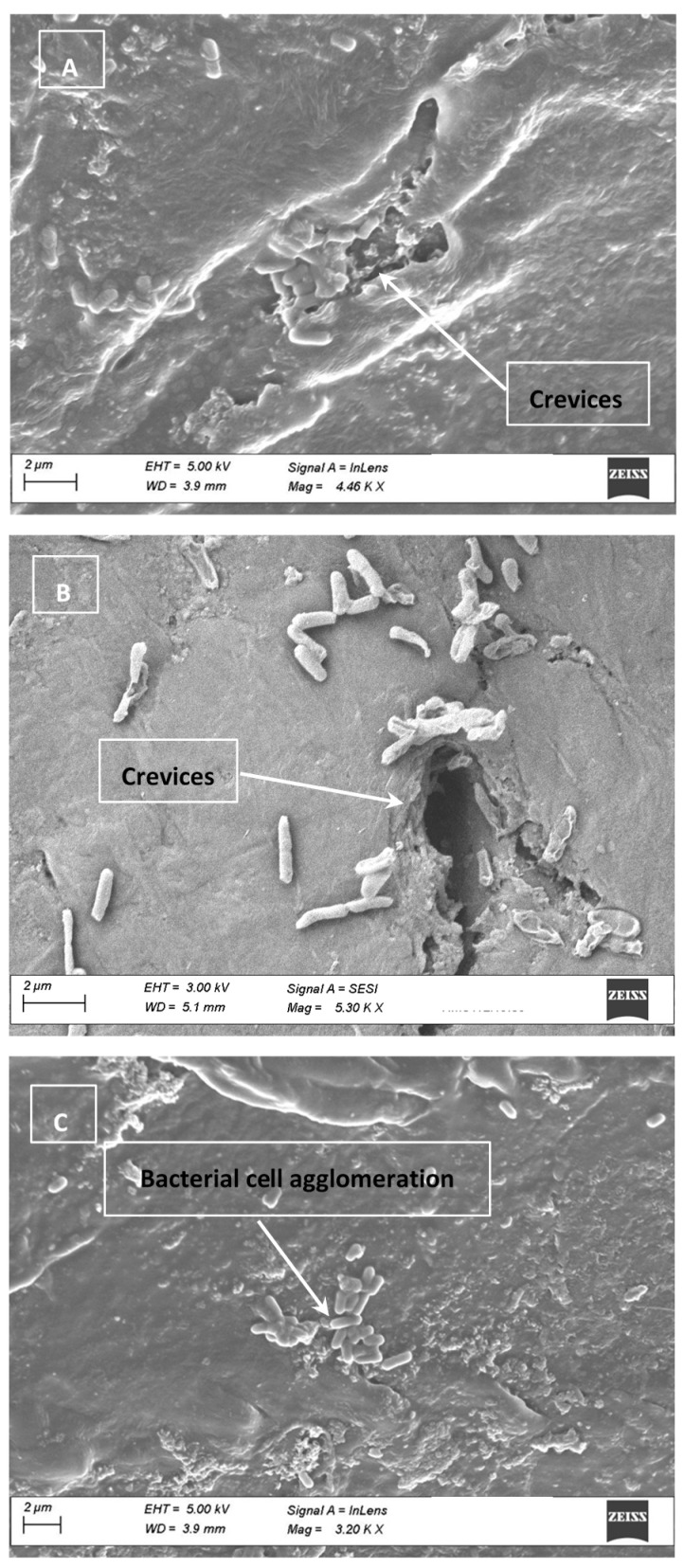
(**A**–**C**) Electron micrograph of three different high-density polyethylene (HD) samples used during the experiments. Based on the scale indicated in the micrograph, the crevice highlighted in (**B**) appears to be large enough to harbor *Salmonella* cells (rod shaped). (**C**) depicts bacterial cell agglomerates present on the surface of the HD.

**Figure 4 foods-10-01459-f004:**
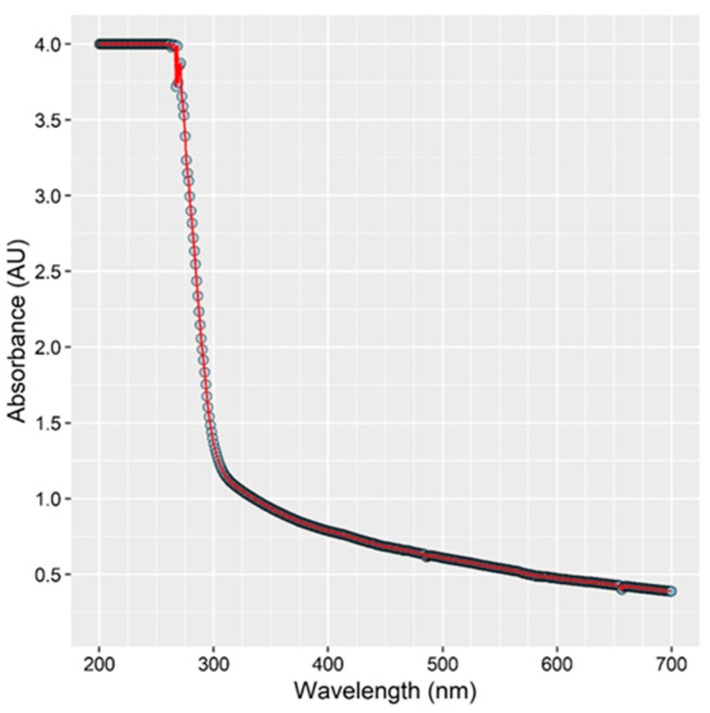
Absorption spectrum of fluid removed from surface of chicken breast (CB). The graph represents the light absorbance of the chicken fluid vs. the wavelength of light. As observed, below 300 nm, the absorption of light increases, which could potentially shelter *Salmonella* and prevent deactivation.

**Table 1 foods-10-01459-t001:** *Salmonella* reduction on chicken breast.

Illumination Time (s)	Irradiance ^1^ (mW/cm^2^)	UV ^4^ Dose (J/cm^2^)	Bacterial Count (Log CFU/cm^2^)	St. Dev. ^5^	Reduction ^2^ (Log CFU/cm^2^)	Bacterial Reduction (%) ^3^
0	2	0	6.21	0.16	-	-
60	2	0.12	5.20	0.48	1.01	90.2
180	2	0.36	4.89	0.81	1.32	95.2
300	2	0.60	4.64	0.67	1.57	97.3
600	2	1.20	4.36	0.70	1.85	98.6
900	2	1.80	4.43	0.70	1.78	98.3
0	4	0	6.26	0.11	-	-
60	4	0.24	4.21	0.77	2.05	99.1
180	4	0.72	3.99	0.86	2.27	99.5
300	4	1.2	3.67	0.63	2.59	99.7
600	4	2.4	3.89	0.44	2.37	99.6
900	4	3.6	3.25	0.53	3.01	99.9

^1^ Irradiance of 2 and 4 mW/cm^2^ are equivalent to 50 and 100%, respectively. ^2^ Reduction based on the initial attachment at time 0. ^3^ Percentage calculated using actual values of colony forming units (CFU) before log transformation. ^4^ Ultraviolet. ^5^ Standard Deviation.

**Table 2 foods-10-01459-t002:** *Salmonella* reduction on stainless steel.

Illumination Time (s)	Irradiance ^1^ (mW/cm^2^)	UV Dose (J/cm^2^)	Log CFU/cm^2^	St. Dev.	Reduction ^2^ (Log CFU/cm^2^)	Bacterial Reduction (%) ^3^
0	2	0	3.4	0.61	-	-
15	2	0.03	2.1	0.72	1.3	93.7
30	2	0.06	1.94	0.83	1.46	95.6
45	2	0.09	1.87	0.72	1.53	96.3
60	2	0.12	1.43	0.41	1.97	98.7
0	4	0	6.27	0.49	-	-
15	4	0.06	4.91	0.56	1.36	95.6
30	4	0.12	3.78	1.5	2.49	99.7
45	4	0.18	3.47	0.65	2.8	99.8
60	4	0.24	2.79	1.76	3.48	99.9

^1^ Irradiance of 2 and 4 mW/cm^2^ are equivalent to 50 and 100%, respectively. ^2^ Reduction based on the initial attachment at time 0. ^3^ Percentage calculated using actual values of colony forming units (CFU) before log transformation. St. Dev. refers to standard deviation and UV refers to ultraviolet.

**Table 3 foods-10-01459-t003:** *Salmonella* reduction on high density polyethylene.

Illumination Time (s)	Irradiance ^1^ (mW/cm^2^)	UV Dose (J/cm^2^)	Log CFU/cm^2^	St. Dev.	Reduction ^2^ (Log CFU/cm^2^)	Bacterial Reduction (%) ^3^
0	2	0	6.58	0.16	-	-
30	2	0.06	5.67	0.28	0.91	87.7
60	2	0.12	5.33	0.12	1.25	94.4
90	2	0.18	5.28	0.17	1.3	95.0
120	2	0.24	5.13	0.16	1.45	96.5
150	2	0.30	5.05	0.25	1.53	97.0
180	2	0.36	4.57	0.47	2.01	99.0
300	2	0.6	2.75	1.68	3.83	99.99
600	2	1.2	2.04	1.68	4.54	99.997
900	2	1.8	1.84	1.46	4.74	99.998
0	4	0	5.20	0.15	-	-
30	4	0.12	3.97	0.24	1.23	94.1
60	4	0.24	3.43	0.24	1.77	98.3
90	4	0.36	2.82	0.29	2.38	99.6
120	4	0.48	2.42	0.07	2.78	99.8
150	4	0.60	2.40	0.00	2.8	99.8
180	4	0.72	0 *	0 *	5.2	99.999
300	4	1.20	0 *	0 *	5.2	99.999
600	4	2.40	0 *	0 *	5.2	99.999
900	4	3.60	0 *	0 *	5.2	99.999

^1^ Irradiance of 2 and 4 mW/cm^2^ are equivalent to 50 and 100%, respectively. ^2^ Reduction based on the initial attachment at time 0. ^3^ Percentage calculated using actual values of colony forming units (CFU) before log transformation. St. Dev. refers to standard deviation and UV refers to ultraviolet. * No colonies recovered.

## Data Availability

Project data has been reported within this manuscript.

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
