# Peer review of "UV-C LED Irradiation Reduces Salmonella on Chicken and Food Contact Surfaces"

_foods, 2021, doi:10.3390/foods10071459_

Round 1

Reviewer 1 Report

The manuscript entitled "UV-C LED irradiation reduces Salmonella on chicken and food contact surfaces" describes the effects of different UV-C LED treatments (time and power) on the survival of adherent Salmonella strains on chicken breast (CB) or surfaces (SS and HD).

The study is well conducted but requires additional information.

Majors concerns:

- Lines 100, 118: Please justify the choice of attachment time (30 min) on the surfaces. Why place the samples in the refrigerator?

- Line 143: How do you detach the bacteria from the surfaces? Are you sure you are removing all adherent cells? Have you estimated the detachment percentage?

- Line 149: Also, is the vortex sufficient to remove all cells on the coupons? Justify your choice because there are other more efficient processes described in the literature (e.g. ultrasound). Couldn't Salmonella be in a viable but non-culturable (VBNC) state?

- Lines 216-230: Couldn't another hypothesis be the increased resistance capacity of adherent bacteria subjected to stresses compared to bacteria in vegetative state as it has already been widely shown in particular for Listeria monocytogenes? 

Minors concerns:

- In the introduction, the state of the art on the use of light in food to reduce the microbial load is not described.

- Lines 75-76: Please describe how you prepare the cocktail from different Salmonella strains. How do you make sure they are in the same proportions? Do you perform an enumeration?

- Lines 121: Please justify your choice to stop at 60 s for SS while for CB and HD you test up to 900 s.

- Table 3: Please explain why you only got an average of 3.4 log CFU/cm2 of Salmonella subjected to 2mW/cm at T0 on SS?  And the impact on the results?

Author Response

Point 1: Lines 100, 118: Please justify the choice of attachment time (30 min) on the surfaces. Why place the samples in the refrigerator?

Response 1: Allowing bacterial cultures to attach to inoculated surfaces is a common protocol in microbiology. This prevents droplets of the liquid (culture grown in broth) to run-off from the surface.  The attachment time facilitates the inoculum which is in broth (diluent) to dry out.  Doing this at refrigeration also prevents the inoculum to multiply. For example, Chun et al (2010) described in their methods after inoculation of chicken breast: The inoculated chicken breasts were then air-dried in a laminar flow hood for 20 min.” Other publications also indicate the practice of inoculating and allowing certain time for attachment (Calle et al. 2015, Phebus et al. 2000, Woodling and Moraru 2005).

Point 2: Line 143: How do you detach the bacteria from the surfaces? Are you sure you are removing all adherent cells? Have you estimated the detachment percentage?

Response 2: As explained in“Microbial Analysis” within the Methods section of our manuscript, upon completion of each treatment time, inoculated Stainless Steel and High-Density Polyethylene coupons were transferred to sterile 10-ml phosphate-buffered saline (PBS) solution, mixed using a vortex for 60 sec to remove or transfer the bacterial cells from the surface to the saline solution.  The saline solution was plated on the corresponding selective agar to enumerate Salmonella. Same steps were followed with the inoculated untreated control, thus same removal procedure was applied to all surfaces. If detachment was a problem, it would also occur for the control sample.  Since we compared results for treatments with control, any loss due to poor detachment should cancel / be accounted for mathematically. 

This procedure to remove cells from a surface is somewhat common. Nonetheless, we reviewed literature to make sure we were using a known procedure. Usually, researchers working on biofilm formation on same type of surfaces follow similar detachment steps as we used, with the addition of glass beads to detach biofilms, such as the case of Bae et al. (2012). In their methods they describe the following: “For enumerating surviving pathogens on the surfaces of stainless steel coupons, coupons were transferred to 50 ml conical centrifuge tubes containing30 ml of sterile PBS and 3 g of glass beads (425 to 600μm in diameter; Sigma-Aldrich, St. Louis, Mo., USA). Tubes were vortexed (model No.M37610-33, Barnstead International, Dubuque, IA, USA) at maximum speed for 1 min to dislodge and disperse cells from the surfaces of the coupons.” A similar procedure was used by Zhu et al. (2012) in their investigation of the antimicrobial activity of copper against Salmonella. We did not use glass beads since 30 min of attachment under refrigeration don’t facilitate biofilm formation and it was not our intention. Andrade et al. (1998) evaluated effect of sanitizers to reduce bacterial cells inoculated on stainless steel. In their methods, they describe the use of “Plate Count Method” to enumerate microorganisms after applying their treatments, by immersing stainless steel coupons in tubes containing 2 ml of PBS and vortex for 2 min prior plating.

In summary, we followed methods previously used by other studies and corrected potential detachment problems by using the same procedure in control (untreated) and treated samples.

Point 3: Line 149: Also, is the vortex sufficient to remove all cells on the coupons? Justify your choice because there are other more efficient processes described in the literature (e.g. ultrasound). Couldn't Salmonella be in a viable but non-culturable (VBNC) state?

Response 3: As mentioned above, the protocol we have used has successfully been employed in previous literature.  The reviewer suggests alternate procedures (ultrasound).  Such treatments may be effective at detaching cells from surfaces. However, given the controls were coupons which were treated similarly to samples irradiated, any losses would be expected to mathematically cancel.  We have no evidence to suggest differences in bacterial detachment between control and treated samples, and therefore feel the results are reliable.

It may also be possible to have some bacterial cells that, as a result of the treatment were injured and could be non-culturable. However, that was not explored.  We did not intend to resuscitate Salmonella subjected to UV treatment, which rather requires more complex and extended procedures (Gupte et al. 2003).

Point 4: Lines 216-230: Couldn't another hypothesis be the increased resistance capacity of adherent bacteria subjected to stresses compared to bacteria in vegetative state as it has already been widely shown in particular for Listeria monocytogenes? 

Response 4: It is possible to have an additional hypothesis explaining why Salmonella had a lower destruction rate in chicken. As indicated by the reviewer there is a possibility of increased resistance to UV irradiation caused by exposure to various stressors. There is also a possibility of repair when certain genes are activated. During our literature review we identified the research by Estilo and Gabriel (2017). They observed resistance of Salmonella to UV after exposure of the microorganism to acid, desiccation, heat, and combinations of those stressors. 

However, in our study we found that the efficacy of UV to kill Salmonella was reduced only in the case of chicken breast. In other words, the reduction rate was different between food contact surfaces and the chicken.   Due to those findings, we did not want to consider that the lowered reduction of Salmonella on chicken breast was potentially associated with resistance of adherent cells subjected to stress.

Point 5: In the introduction, the state of the art on the use of light in food to reduce the microbial load is not described.

Response 5: We have added a paragraph in the introduction describing current use of UV light in the food industry, adding two new references. This new information was inserted in the second paragraph of the introduction.

Point 6: Lines 75-76: Please describe how you prepare the cocktail from different Salmonella strains. How do you make sure they are in the same proportions? Do you perform an enumeration?

Response 6: We would like to respond to this comment in three parts. i)The procedure to prepare the cocktail from different Salmonella strains is provided in Methods, section Bacterial cultures.  The text as currently appears in the manuscript is pasted below with some sections underlined to highlighting information.  As indicated, we used five different strains that were grown overnight individually (they were originally frozen at -80°C).  ii)We made sure to use always 2ml from each culture, making a total of 10 ml of the bacterial cocktail. Our pipettes are subjected to frequent calibration as per our lab policies. iii)We did enumerate as indicated in the text below.

A five-strain Salmonella cocktail was prepared with Salmonella Thyphimurium ATCC 70 BAA-712, Salmonella Newport ATCC 6962, Salmonella Enteritidis ATCC 31194, Salmonella 71 Senftenberg ATCC 43845, and Salmonella Heidelberg ATCC 8326. Each strain was grown 72 individually by transferring 10 µl from the stock culture into 9-ml of Tryptic Soy Broth 73 (TSB) (EMD Millipore Chemicals; Darmstadt, Germany) and incubating for 24 h at 37°C. 74 Equal amounts from each grown Salmonella suspension were combined into a sterile test 75 tube and homogenized. The bacterial cocktail was freshly prepared prior each repetition. 76 Salmonella concentration in the cocktail was confirmed at each repetition of the experiment 77 by conducting serial dilutions and plating onto Trypticase Soy Agar (TSA) (Becton, Dick-78 inson and Company, France), followed by incubation for 24h at 37°C and subsequent enu-79 meration.

Point 7: Lines 121: Please justify your choice to stop at 60 s for SS while for CB and HD you test up to 900 s.

Response 7: Experiments with stainless steel were the first we conducted. We used shorter exposure times expecting a very efficient and quick destruction of Salmonella with the UV light on SS. Experiments with SS were already conducted, and we didn’t think it was good to discard those results and repeat experiments. However, expecting that extended times would provide higher reductions, we decided to increase the exposure on CB and HD.

Point 8: Table 3: Please explain why you only got an average of 3.4 log CFU/cm2 of Salmonella subjected to 2mW/cm at T0 on SS?  And the impact on the results?

Response 8: Consistent with the explanation provided in the previous comment, after we increased exposure time for CB and HD, we considered it was important to increase the target initial inoculation level. That would allow us to observe a larger destruction of Salmonella over time.

References used in the responses (not formatted):

  1. H. Chun, J.Y. Kim, B.D. Lee, D.J. Yu, K.B. Song. Effect of UV-C irradiation on the inactivation of inoculated pathogens and quality of chicken breasts during storage. Food Control, Volume 21, Issue 3, 2010, Pages 276-280,
  2. Calle A, Porto-Fett AC, Shoyer BA, Luchansky JB, Thippareddi H. Microbiological Safety of Commercial Prime Rib Preparation Methods: Thermal Inactivation of Salmonella in Mechanically Tenderized Rib Eye. J Food Prot. 2015 Dec;78(12):2126-35. doi: 10.4315/0362-028X.JFP-15-154. PMID: 26613906.3.
  3. Phebus, R. K., H. Thippareddi, S. Sporing, J. L. Marsden, and C. L. Kastner. 2000. Escherichia coli O157:H7 risk assessment for blade tenderized beef steaks, p. 117–118. In Report of progress 850. Kansas State University, Manhattan.
  4. Woodling, S.E. and Moraru, C.I. (2005), Influence of Surface Topography on the Effectiveness of Pulsed Light Treatment for the Inactivation of Listeria innocua on Stainless-steel Surfaces. Journal of Food Science, 70: m345-m351
  5. Young-Min Bae, Seung-Youb Baek, Sun-Young Lee. Resistance of pathogenic bacteria on the surface of stainless steel depending on attachment form and efficacy of chemical sanitizers. International Journal of Food Microbiology, Volume 153, Issue 3, 2012, Pages 465-473.
  6. NELIO J. ANDRADE, TRACY A. BRIDGEMAN, EDMUND A. ZOTTOLA; Bacteriocidal Activity of Sanitizers against Enterococcus faeciumAttached to Stainless Steel as Determined by Plate Count and Impedance Methods. J Food Prot 1 July 1998; 61 (7): 833–838.
  7. Libin Zhu, Jutta Elguindi, Christopher Rensing, Sadhana Ravishankar,

Antimicrobial activity of different copper alloy surfaces against copper resistant and sensitive Salmonella enterica, Food Microbiology, Volume 30, Issue 1, 2012, Pages 303-310.

  1. Gupte AR, De Rezende CL, Joseph SW. Induction and resuscitation of viable but nonculturable Salmonella enterica serovar typhimurium DT104. Appl Environ Microbiol. 2003;69(11):6669-6675.
  2. Emil Emmanuel C. Estilo, Alonzo A. Gabriel. Previous stress exposures influence subsequent UV-C resistance of Salmonella enterica in coconut liquid endosperm, LWT Food Science and Technology, Volume 86, 2017, Pages 139-147

Reviewer 2 Report

it is an interesting paper showing the effect of the time of treatment, intensity of the radiance, efficiency depending of the type of surface and also the absorption of the UV C through the liquid drop from the poultry. I would also be interested to see what is the efficiency of the UV C through the packaing film used to pack the poultry.

I have corrections to propose

in the abstract, add the log inhibition with the 99.999% reduction

when data are significant it should be p   0.05

section I suggest to add the information about were the Salmonella used were  isolated

section 2.6 add information about why this analysis was one

the text in the conclusion is in realty a discussion: put this text in the discussion

In the conclusion add a summary of the effectiveness of the UV C treatment in relation with the time of treatment; intensity of radiance; dose of irradiation and type of surface.

Author Response

Point 1: In the abstract, add the log inhibition with the 99.999% reduction.

Response 1: The log inhibition was added as requested. The updated sentence reads: “Longer exposure times of HD to UV-C yielded up to 99.999% (5.0 Log CFU/cm2) reduction of Salmonella with both irradiance levels.”

Point 2: When data are significant it should be p <   0.05. 

Response 2: Change was made throughout the document.

Point 3: Section I suggest to add the information about were the Salmonella used were isolated

Response 3: Strains were all obtained from ATCC and currently are part of the bacterial collection of the International Center for Food Industry Excellence (ICFIE) at Texas Tech University.  We intended to add the information as requested by the reviewer. Unfortunately, ATCC does not provide isolation information for all strains. For example, one of the strains is regarded as “Patent deposit”. According to the ATCC website: “ATCC Technical Services does not have technical information on patent deposits that are not produced or characterized by ATCC.” For other strains, technical ATCC information indicates “Provided AS IS”.

Point 4: section 2.6 add information about why this analysis was done

 Response 4: In section 2.6 we added a sentence at the beginning of the paragraph that says “Fluids associated with CB were analyzed to evaluate whether they could offer a protective coat effect for bacteria by absorbing ultraviolet light. The extent of light absorption by CB juices was studied by ultraviolet-visible (UV-VIS) absorption spectroscopy.”

To complement this explanation for the reviewer, we would like to mention that when analyzing experimental results, we noted that efficacy of UV light on chicken breast was not comparable to food contact surfaces.  While most Salmonella was inactivated quickly during UV treatment, there existed a population of cells which seemed to be unaffected by further UV dose.  We want to understand what mechanism is impeding further Salmonella destruction and stated three hypotheses for this (cell clumping; sheltering in crevices; and light absorption by other sample components).  The chicken wash fluid was analyzed to provide supporting evidence for the possibility that fluids associated with CB can coat and immerse bacterial cells – thus protecting them from UV irradiation.

Point 5: the text in the conclusion is in realty a discussion: put this text in the discussion

Response 5: The first paragraph in “Conclusions” was removed. We put it in “Results and Discussion” and is now the last paragraph of that section.

Point 6: In the conclusion add a summary of the effectiveness of the UV C treatment in relation with the time of treatment; intensity of radiance; dose of irradiation and type of surface.

Response 6: We have added a paragraph summarizing as requested by the reviewer. The paragraph appears in the conclusion and reads: “By doubling the irradiance (mW/cm2) from 50 to 100%, the UV dose (J/cm2) deposited on each surface was also increased or duplicated. Larger UV doses were directly correlated to the Salmonella reduction (Log CFU/cm2) attained on each surface tested; however, such reduction did not necessarily double. In other words, Salmonella reductions were consistent with the intensity of exposure but not exactly proportional to the increase in the UV dose.”

Round 2

Reviewer 1 Report

Thanks for your answers which correspond to my expectations. I have no further comments on your publication.

Reviewer 2 Report

the corrections were done as requested